# Ploidy in *Vibrio natriegens*: Very Dynamic and Rapidly Changing Copy Numbers of Both Chromosomes

**DOI:** 10.3390/genes14071437

**Published:** 2023-07-13

**Authors:** Patrik Brück, Daniel Wasser, Jörg Soppa

**Affiliations:** Institute for Molecular Biosciences, Goethe University, Max-von-Laue-Str. 9, D-60438 Frankfurt, Germany

**Keywords:** *Vibrio natriegens*, ploidy, polyploidy, chromosome copy number, cell size, cell volume, origin of replication, terminus of replication, growth curve, dynamic regulation

## Abstract

*Vibrio natriegens* is the fastest-growing bacterium, with a doubling time of approximately 12–14 min. It has a high potential for basic research and biotechnological applications, e.g., it can be used for the cell-free production of (labeled) heterologous proteins, for synthetic biological applications, and for the production of various compounds. However, the ploidy level in *V. natriegens* remains unknown. At nine time points throughout the growth curve, we analyzed the numbers of origins and termini of both chromosomes with qPCR and the relative abundances of all genomic sites with marker frequency analyses. During the lag phase until early exponential growth, the origin copy number and origin/terminus ratio of chromosome 1 increased severalfold, but the increase was lower for chromosome 2. This increase was paralleled by an increase in cell volume. During the exponential phase, the origin/terminus ratio and cell volume decreased again. This highly dynamic and fast regulation has not yet been described for any other species. In this study, the gene dosage increase in origin-adjacent genes during the lag phase is discussed together with the nonrandom distribution of genes on the chromosomes of *V. natriegens*. Taken together, the results of this study provide the first comprehensive overview of the chromosome dynamics in *V. natriegens* and will guide the optimization of molecular biological characterization and biotechnological applications.

## 1. Introduction

For many years, it has been thought that archaea and bacteria (prokaryotes) have a single copy of a circular chromosome and, thus, that they are monoploid (often called haploid). However, during the last decade, it has been revealed that many prokaryotic species are oligoploid (2–10 copies), polyploid (more than 10 copies), or even hyperpolyploid (more than 100 copies). This is true for various groups of prokaryotes, e.g., proteobacteria [1], cyanobacteria [2], Firmicutes [3], halophilic Archaea [4], and methanogenic Archaea [5]. In addition, all very large bacteria (“giant bacteria”) seem to be polyploid, e.g., *Epulopiscium*, *Achromatium oxaliferum*, and large sulfur bacteria [6,7]. In stark contrast with earlier beliefs, it now seems that the majority of prokaryotic species are oligo-/polyploid and that truly monoploid species, such as *Caulobacter crescentus*, *Wolinella succinogenes*, and *Synechococcus* sp. WH 8101, comprise the minority [1,2]. Many different advantages of polyploidy for prokaryotes have been theoretically discussed or experimentally verified, including a very low mutation rate, resistance against double-strand breaks, and the ability to grow in the absence of any environmental phosphate source [8,9]. Polyploidy is not restricted to prokaryotes; it has been known for a long time that many species of eukaryotes are polyploid, especially in plants, ciliates, amphibians, and fish but also other groups [10,11,12].

Ploidy level is not a defining trait for specific phylogenetic groups, and indeed, species with very different ploidy levels occur in many phylogenetic groups. Therefore, it has been concluded that oligo-/polyploidy evolved in different groups of prokaryotes independently at different times and for different reasons. For example, Gammaproteobacteria include the following species with very different ploidy levels: (1) *Escherichia coli*, which is monoploid during very slow growth and mero-oligoploid with 6.8 origins and 1.7 termini during fast growth [1,13]; (2) *Pseudomonas putida* and *Zymomonas mobilis*, which are polyploid with approximately 20 origins and 10–16 termini [1,14]; and (3) *Buchnera* sp., which is hyperpolyploid with 120 copies of the chromosome [15]. The chromosome copy number of another gammaproteobacterium, *V. natriegens*, remains unknown.

*V. natriegens* has received very rapidly growing attention in recent years. It was isolated from salt marsh mud on Sapelo Island (GA, USA) more than 60 years ago and was first named *Pseudomonas natriegens* (natriegens signifies sodium needing) [16]. It was later renamed *V. natriegens* [17]. Soon after its discovery, it was observed to be able to grow very fast, and a generation time of 9.8 min under optimal conditions has been reported [18]. However, this was only seen when a small inoculum was used, and it was only true for the 15 min of fastest growth; otherwise, a value of 14.1 min was calculated [18]. Similar values of approximately 14 min have been reported in many publications since, making *V. natriegens* the fastest-growing bacterium known to date.

*V. natriegenes* forms the core of the genus together with additional species such as *V. harveyi* and *V. parahaemolyticus* [19]. The genome sequence of the type strain DSMZ 759 became available a decade ago, and it was revealed that *V. natriegens* contains two circular chromosomes of different sizes [20]. It has been hypothesized that the fast growth is based on the large number of 11 rRNA operons and the high rRNA promoter strength [21].

*V. natriegens* has a very versatile metabolism and can grow aerobically and anaerobically on a large number of carbon sources. It is ideal for molecular biological applications because it can take up exogenous DNA and incorporate it into its genome. *V. natriegens* is compatible with *E. coli* vectors, and CRISPR applications have been established, including workflows that lead to the death of nonedited cells, enable large-scale interspecies gene transfer, and allow for multiplex genome editing [22,23,24,25]. Notably, a genome-wide insertion screen identified genes that are important or essential for growth in a complex medium or in a synthetic medium with glucose or fructose [26]. Several synthetic biology approaches have been described [27,28], including the generation of a Golden Gate cloning toolbox with 191 genetic parts [29]. *V. natriegens* has been used for the production of heterologous proteins, both in vivo and using a cell-free in vitro system. Notably, this includes the production of therapeutic proteins and membrane proteins as well as the incorporation of non-natural amino acids [30,31,32,33,34,35,36,37,38,39,40]. Together, these features of *V. natriegens* make it interesting as a chassis for various biotechnological applications [41,42]. For example, it has been used to produce 1,3- and 2,2-propanediol [43,44], succinate [45] several amino acids [46], l-DOPA [47], melanin [48], and various proteins (see above).

For basic molecular genetic research as well as biotechnological applications, knowledge of the chromosome copy number of *V. natriegens* appears to be important. Another species of the genus, *V. cholerae*, has been intensively studied and described to be monoploid. When an origin-adjacent site was (indirectly) tagged with a fluorescent protein, one focus (prior to replication) or two foci (after replication) were observed, which were transported to the cell poles prior to cell division [49,50,51]. In stark contrast, one publication claimed that *V. cholerae* has a variable number of chromosomes, with 56–72 during the early stationary phase [52]. In addition, several other Gammaproteobacteria are not monoploid but oligoploid or polyploid [1].

In the present study, we aimed to close this knowledge gap and quantify the chromosome copy number of *V. natriegens*. The well-established real-time PCR method was optimized for application to *V. natriegens* and used for the quantification of the number of replication origins as well as the number of termini of both chromosomes throughout the growth curve. As an independent method, marker frequency analyses were also performed throughout the growth curve. In addition, the cell volume was determined, and the numbers of origins/termini per unit volume were calculated. It was found that the copy number is highly regulated and that this regulation differs from that of the close relative *V. cholerae*.

## 2. Materials and Methods

### 2.1. Strain, Medium, and Growth Curves

The *V. natriegens* type strain DSM 759 was obtained from the German Collection of Microorganisms and Cell cultures (www.dsmz.de, accessed on 1 February 2022). It was grown in a complex medium (10 g/L tryptone, 5 g/L yeast extract, and 25 g/L NaCl), as described at www.biorxiv.org/content/10.1101/775437v1 (accessed on 1 February 2022). In brief, 30 mL cultures in 100 mL Erlenmeyer flasks were grown at 37 °C with rapid shaking at 200 rpm in a rotary shaker. The test cultures were inoculated with 24 h stationary-phase cultures with a starting cell density of 3 × 10^7^ cells/mL. Aliquots were removed at the time points indicated in the figures, and cell density was determined using a Neubauer counting chamber (0.02 mm depth). At the time points selected for copy number quantification or marker frequency analysis (see Figures), aliquots of approximately 2–6 × 10^8^ cells were removed, and the cells were collected via centrifugation. The supernatants were also removed, and the cell pellets were frozen at −80 °C until analysis.

### 2.2. Quantification of Genome Copy Numbers Using Real-Time PCR

Figure 1 gives an overview of the different steps of the real-time PCR method for the quantification of different sites of prokaryotic chromosomes. The different steps of the method are described below.

#### 2.2.1. Generation of Standard Curves

*V. natriegens* was grown as described above, and genomic DNA was isolated using a standard procedure and used as a template for the amplification of approximately 1 kb fragments of regions of chromosomes 1 and 2 near the replication origin (at 0% on the chromosomal map), at 17%, at 33%, and near the replication terminus (at 50%). The oligonucleotides used for the amplification of the eight fragments are listed in Appendix A. Analytical agarose gels were used to verify that in each case, only a single fragment of the expected size was amplified. The fragments were purified using the GenElute PCR Clean-Up kit obtained from Sigma-Aldrich (St. Louis, MO, USA), and the spectra of the isolated fragments were recorded using a Nanodrop ND-1000. Mass concentrations were calculated by assuming that the OD_260_ represents a DNA concentration of 50 ng/mL, and the molar concentrations were determined using the molecular weights of the individual fragments determined with the DNA molecular weight calculator (https://www.bioinformatics.org/sms2/dna_mg.html, accessed on 1 February 2022). The number of PCR fragment molecules per volume was calculated using Avogadro’s number (6.023 × 10^23^ molecules per mol). Dilutions from 10^4^ to 10^8^ of the standard fragments were analyzed using real-time PCR, and the results were used to generate standard curves for each fragment.

#### 2.2.2. Cell Disruption

Cell pellets were thawed and resuspended in 1 mL of TE buffer (10 mM Tris/HCl pH 8.0, 2.5 mM EDTA, and 20 mM NaCl). The cell suspension was transferred to a 2 mL Eppendorf tube containing 0.5 g silica beads (0.1 mm, Roth, Karlsruhe, Germany). Cell disruption was performed using a speed mill with 8 cycles of 30 s each at 4 °C and intermediate cooling. The beads and cell debris were removed via centrifugation, and different dilutions of the cytoplasmic extract were used for real-time PCR analysis without further treatment.

The method described above was superior to alternative methods of cell disruption, e.g., treatment with 2 mg/mL of lysozyme at 37 °C for one hour or treatment of the cell extract with Chelex 100 to remove metal ions after cell disruption with a speed mill.

It was verified that further treatment of the cytoplasmic extract was not necessary because the dilutions of the extract led to the theoretically predicted shift in C_T_ values that occur for strictly exponential amplifications of PCR fragments.

#### 2.2.3. Quantification of Copy Numbers of Different Chromosomal Sites

The dilutions of the standard fragments (10^4^–10^8^-fold) and cytoplasmic extracts (10-/100-/500-fold) were analyzed simultaneously using real-time PCR. The sizes of the amplified analysis fragments were approximately 300 nt, and the sequences of all oligonucleotides are listed in Appendix A. Real-time PCR was performed using a Rotor-Gene 3000. SYBR Green for real-time PCR was obtained from Lumiprobe (Hannover, Germany). The following program was used: (1) 10 min at 96 °C, (2) 40 cycles of 30 s at 96 °C, 30 s at 5 °C below the melting temperature of the respective oligonucleotides, 30 s at 72 °C, and (3) 5 min at 72 °C. Both melting temperature analysis and analytical agarose gels were used to guarantee that only PCR fragments of the selected sites were amplified. For each dilution of each sample, two technical replicates were used for each of the three biological replicates that were analyzed, such that the results are based on a multitude of replicates. The C_T_ values of the standard samples were used to generate standard curves (no. molecules versus C_T_ values), which were used to determine the number of molecules in the dilutions of the cytoplasmic extracts. Together with the number of cells included in the analysis, the average number of sites per cell and their standard deviation were calculated.

### 2.3. Quantification of Cell Volumes

Aliquots for the quantification of chromosome copy numbers and determination of cell volumes were taken simultaneously. An Axioskop 40 light microscope (Zeiss, Jena, Germany) was used to visualize the cells. Images were taken using an AxioCam MRm camera, and cell lengths and diameters were determined using AxioVision software (release 6). The dimensions of 25 cells were determined for each replicate. Average values were determined and used to calculate average cell volumes using the geometrical formula for cylinders (V = π r^2^ h). It should be noted that this formula was used for all time points, even if the cells became much shorter and the population comprised short rods and spheres during the stationary phase. However, the difference in the formulas for cylinders and spheres (V = 4/3 π r^3^) is only 30% and thus much smaller than the cell size differences that were observed throughout the growth curve.

### 2.4. Bioinformatic Analyses

The genome sequence of *V. natriegens* was retrieved from https://www.ncbi.nlm.nih.gov/assembly/GCF_001456255.1 (accessed on 1 February 2022). It was loaded into Clone Manager Professional Suite 8. The design of all oligonucleotides was performed with Clone Manager. The melting points of the oligonucleotides were calculated at https://tmcalculator.neb.com/#!/main (accessed on 1 February 2022). The growth rate µ was calculated using the application “growthcurver” in the R package and was transformed into the doubling time using the formula t_d_ = ln2/µ.

### 2.5. Marker Frequency Analysis

Cells were grown as described above, and aliquots were removed at the time points indicated in the figures. Genomic DNA (gDNA) was isolated using the GeneJET Genomic DNA Purification Kit (Thermo Scientific, Waltham, MA, USA) according to the manufacturer’s instructions. The samples were sent to Novogene (www.novogene.com, accessed on 1 February 2022) for next-generation Illumina sequencing (2 × 150 nt paired-end reads), which resulted in 7 to 10 million reads per sample. The sequences were analyzed using the galaxy platform (https://usegalaxy.eu/, accessed on 1 February 2022). The “FastQC” tool (Galaxy Version 0.73+galaxy0) was used for quality control, and then the “RNA STAR” tool (Galaxy Version 2.7.10b+galaxy3) was applied for the mapping of the sequencing reads to the *V. natriegens* reference genome in a BAM file. Standard parameters were applied, except for the following: “Maximum ratio of mismatches to mapped length”, which was set to 0.04; “Minimum alignment score, normalized to read length” and “Minimum number of matched bases, normalized to read length”, which were set to 0.66, respectively, resulting in over 95% uniquely mapped reads for all nine samples. Afterward, the tool “bamCoverage” (Galaxy Version 3.5.1.0.0) was used to create bigWig files without normalization. Final visualization was performed with Integrated Genome Browser [53]. To determine the ratio between the origin and terminus at each time point, the “Bin size in bases” parameter was changed from 1 to 10,000 during BAM coverage for a more representative value of the origin and terminus region. The value of the respective bin was then used for calculation.

## 3. Results

### 3.1. Optimization of the Real-Time PCR Method for Application to V. natriegens

Figure 1 gives a schematic overview of the real-time PCR (qPCR) method for the quantification of chromosome copy numbers. It offers the advantage of being able to independently quantify the copy numbers of different chromosomes and/or different sites, and in the current study, we made use of both possibilities. Standard PCR was used to amplify the region of interest, and then the concentration of the purified PCR fragment was determined (molecules per volume) and used to generate a standard curve. A known number of cells were disrupted, and a dilution series of the cytoplasmic extract was analyzed using real-time PCR along with the standard curve. The results enabled us to calculate the copy number per cell. In particular, two steps of the procedure needed to be reoptimized for each species under investigation. In our case, the *V. natriegens* type culture DSM 759 was obtained from the German Collection of Microorganisms and Cell Cultures (www.dsmz.de, accessed on 1 February 2022). First, the cell disruption method guarantees that much more than 90% of the cells are lysed and that the genomic DNA remains mostly intact. Several cell disruption procedures were tested, and the use of a speed mill (eight cycles) was superior; in contrast, lysozyme treatment was very inefficient. Second, it was verified that the cytoplasmic extract of *V. natriegens* did not contain any substances that would decrease the efficiency of PCR; therefore, dilutions of the cytoplasmic extract were used without any further treatment. The method was then applied for the quantification of the copy numbers of the replication origins and replication termini of both chromosomes plus two additional sites on chromosome 1. For each site, two forward and two reverse oligonucleotides were generated, and four possible combinations were tested. Oligonucleotide pairs that yielded single products with exponential amplification were selected (the oligonucleotides are listed in Appendix A).

### 3.2. Quantification of the Copy Numbers of Origins and Termini of Both Chromosomes throughout the Growth Curve

*V. natriegens* (type culture DSM 759) was grown in a complex medium under optimal conditions. Aliquots were removed and analyzed at 14 time points throughout the growth curve, representing the lag phase, exponential phase, and stationary phase. Cell densities were determined by counting using a Neubauer counting chamber, and a growth curve was derived. The doubling time during the exponential phase (1–3 h post-inoculation) was 14.7 min. The copy numbers of chromosomal sites were quantified using qPCR at nine selected time points. The results are shown in Figure 2A for chromosome 1 and in Figure 2B for chromosome 2. A very high increase in the copy number was observed for the origins of both chromosomes at the end of the lag phase, with a maximum at the start of the exponential phase (1.5 h post-inoculation). The origin copy numbers per cell rapidly decreased during the exponential phase and were low and constant from the early to late stationary phase. The dynamic increase and decrease were identical for both origin regions, but the absolute numbers differed considerably. While the copy number of origin 1 increased to 51.6, the copy number of origin 2 increased to “only” 21.3 (Table 1). In stark contrast with the origins, the number of the two termini did not vary much throughout the growth curve. Taken together, there was high variation in the origin/terminus ratio (ore/ter ratio) (Table 1), meaning that the dosage of genes near origin 1 exceeded that near terminus 1 more than 6-fold during the early exponential phase, and the difference in the stationary phase was much less than 2-fold (2.5-fold and 60% for the origin/terminus ratio of chromosome 2).

In summary, *V. natriegens* is mero-polyploid with regard to both chromosomes in the early exponential phase but oligoploid in the stationary phase.

### 3.3. Quantification of the Copy Numbers of Additional Chromosomal Sites

The large differences observed between the copy numbers of the replication origins (location at 0% of the chromosome) and the termini (at 50% of the chromosome) prompted us to quantify the copy numbers of two additional sites, at 17% and 33% of the chromosome and equidistant between the 0% and 50% sites. Copy numbers were quantified at 1.5 h after inoculation to represent the highest origin copy number and at 24 h after inoculation to represent the late stationary phase. The results are shown in Figure 3. If replication were to occur in a steady state and proceed with linear speed, copy numbers of 1/3 and 2/3 of the copy number differences between the origins and termini would be expected, respectively. For the origin copy numbers of chromosome 1 at the early exponential phase, the expected values for the 17% and 33% sites were 35.5 and 19.5, respectively. In stark contrast, the experimentally determined values were much lower (Figure 3). The same was true for chromosome 2: the expected values for the 17% and 33% sites were 15.3 and 9.2, but the experimentally determined values were considerably lower. These differences clearly show that at the onset of exponential growth, the replication of *V. natriegens* is not in a steady state, even though many rounds of replication have been initiated, multiplying the genes near the origin but not yet reaching the 17% and 33% locations of the chromosome. At 1.5 h after inoculation, there was no linear gene dosage difference along the chromosome, but there was a specific several-fold increase in the gene dosage for genes very close to the replication origin.

Unexpectedly, specific gene dosage enrichment was also observed in the late stationary phase when replication should have long ceased. For both chromosomes, the gene dosage near the origin was higher than the gene dosage at the 17%, 33%, and 50% sites (terminus) of the chromosome, with the same copy number. However, this was not observed using marker frequency analysis (see below).

### 3.4. Quantification of the Cell Volume and the Copy Numbers per Unit Volume

During cell counting, it became obvious that the cell volume changed drastically, e.g., cells were much larger during the early exponential phase than during the late stationary phase (Figure 4A,B). Therefore, the average cell volume was determined at all 14 time points throughout the growth phase. The cell volume sharply increased during the late lag phase and peaked at the onset of exponential growth 1.5 h after inoculation (Figure 4C). During exponential growth, the cells divided faster than the cell volume doubled, and therefore, the cell volume rapidly decreased again. To enable a comparison of the dynamic chromosome copy number changes with the dynamic cell volume changes, the origin and terminus copy numbers per unit cell volume were calculated (Figure 4C,D). The cell volume increase outperformed the copy number increase, such that the copy numbers per µm^3^ cell volume decreased in the late lag phase and early exponential phase, further decreasing during the exponential phase due to the very rapid decrease in the average cell volume. From the early to late stationary phase, the copy number per unit cell volume increased again because the chromosome copy number stayed constant (Figure 2), and the cell volume further decreased (Figure 4).

### 3.5. Marker Frequency Analysis

The dynamic changes in the origin numbers and origin/terminus ratios that were observed were unexpected. Therefore, we aimed to verify these results using a second, independent method. Marker frequency analysis was chosen because (1) it does not involve an amplification step, (2) it generates relative abundances of all nucleotides around both chromosomes, (3) it does not require any manipulations of the cells prior to harvesting and DNA isolation, and (4) it is another well-established method. To this end, a *V. natriegens* culture was grown, and samples were removed at the same time points as those used for the qPCR analysis described above. Genomic DNA was isolated using a genomic DNA isolation kit (Thermo Scientific, Waltham, MA, USA), and next-generation sequencing was performed by Novogene (www.Novogene.com, accessed on 1 February 2022). Between 7.15 and 10.08 million paired-end reads of 150 nt were generated and mapped to the genome of *V. natriegens*. The average was 8.27 million reads, which corresponds to an average coverage of 308 (the range for all samples was 266–375). The reads were mapped onto the *V. natriegens* genome and visualized without normalization (Figure 5). At the beginning of the experiment, there was an equal distribution of reads throughout both chromosomes. However, 30 min after inoculation, the fraction of reads around the origin of replication of chromosome 1 increased, indicating that replication started very early in the lag phase, long before the onset of cell division. At one-hour post-inoculation, the fraction of reads around the origin had further increased, and a continuous gradient between the origin and the terminus region had developed, indicating the progression of replication bidirectionally along both arms of chromosome 1. This picture was similar at the next three time points from 1.5 h to 2.5 h post-inoculation. In stark contrast, replication had nearly completely ceased at 4 h, at the end of the transition phase. No replication occurred at 5.5 h and 24 h, i.e., during the stationary phase.

It has been reported that the replication of the *crtS* locus on chromosome 1 licenses replication initiation on chromosome 2; thus, chr2 initiates later than chr1 [50,51]. Our marker frequency analyses (Figure 5) confirmed this view. At 30 min post-inoculation, there was no indication of replication initiation for chromosome 2, in contrast with chromosome 1. Ongoing replication of chromosome 2 was observed at the four time points from 1 h to 2.5 h post-inoculation. After 4 h, replication had nearly ceased, and no replication of chromosome 2 was observed at 5.5 h and 24 h, which was similar to chromosome 1.

## 4. Discussion

In this study, two independent experimental approaches were used to study chromosome dynamics in *V. natriegens* throughout the growth phase, from the inoculation of fresh medium with stationary-phase cells to the lag phase, exponential phase, transition phase, and stationary phase. Both methods have specific advantages, i.e., with the qPCR method, it is possible to quantify the absolute numbers of analyzed sites per cell, whereas marker frequency analysis provides an overview of the whole genome.

Both methods revealed that the replication of chromosome 1 started 30 min after inoculation, approximately 1 hour before the cells started to divide. To facilitate the comparison of the results of both methods, the origin/terminus ratios were calculated and are shown for the qPCR method (Figure 6A) and for marker frequency analysis (Figure 6B). For both methods, the highest values were at the end of the lag phase and the onset of exponential growth. The peaks showed a slight time difference, being at 1.5 h for the qPCR method and 2 h for marker frequency analysis. Notably, both methods revealed a very dynamic change in the ori/ter ratio, with a rather narrow peak around the onset of exponential growth and much lower values before (0.5 h) and after (4 h) this point. To our knowledge, this is the first time that the chromosome dynamics of *V. natriegens* have been analyzed throughout the growth curve, though it is typical for a snapshot of the cells in the mid-exponential growth phase to be generated. Another common result of both methods was that the ori/te ratio for chromosome 1 exceeded the value of that for chromosome 2, clearly showing that multifork replication occurs.

Multifork replication was first described to occur in fast-growing cultures of *E. coli* and *Bacillus subtilis* [54,55]. During slow growth, when the doubling time is longer than the time required to replicate and segregate the chromosome, monoploid prokaryotes contain one copy of the chromosome in the G1 phase and two copies in the G2 phase and a partly replicated chromosome in the S phase. The origin/terminus ratio is two in replicating cells and one before and after the S phase. When the doubling time is shorter than the time to replicate and segregate the chromosome (approximately 1 h), cells initiate replication before the previous round of replication has terminated, and the replication cycles become intertwined. Replicating chromosomes in fast-growing cells contain four or eight replication origins per terminus when two or three replication cycles become intertwined. This mechanism is called “multifork replication”, and the cells are called “mero-oligoploid”. For fast-growing cultures of *E. coli*, average values of 6.5/6.8 origins and 1.7/2.0 termini have been reported, resulting in an origin/terminus ratio of 4.0/3.3 [1,13]. Similar values have been reported for fast-growing exponential cultures of *B. subtilis*, i.e., 5.9 origins and 1.2 termini, resulting in an origin/terminus ratio of 4.9 [3].

For *V. natriegens*, the highest ori/ter ratio was four with the marker frequency analysis and nine with the qPCR method. Two explanations for this quantitative difference seem possible. First, read numbers vary considerably from base to base (see Figure 5); therefore, a sliding window of 10,000 bases was used to generate the ori/ter ratio graphic shown in Figure 6B. In contrast, the qPCR method used an analysis fragment of only approximately 300 nt. However, the usage of a smaller sliding window of 1000 nt in the marker frequency analysis did not lead to a higher ori/ter ratio. A second explanation is that the qPCR method might have slightly overestimated the origin copy number. In the stationary phase, an ori/ter ratio of 1.4 was determined for chromosome 1 (Table 1), and the number of origins was slightly higher than the copy number of the three other sites (Figure 3B). However, the marker frequency analysis showed that replication ceased in the stationary phase; therefore, all sites around the genome should have an identical copy number. It should be noted that four quantitative values are needed to compute the ori/ter ratio with the qPCR method, i.e., the exact number of molecules in the solutions of the origin standard fragment and the terminus standard fragment, and the exact C_t_ values for both analysis fragments in the qPCR (an exponential analysis). If it is assumed that the ori/ter ratio might have been overestimated by 40% when using the qPCR method and all values are divided by 1.4, the highest ori/ter ratio would decrease from 9 to 6.4, a value similar to the ratio of 4.0 determined using marker frequency analysis. A recent marker frequency analysis of *V. natriegens* determined an ori/ter ratio of approximately five [56], which is between the two values determined in this study. Taken together, three independent analyses showed that the ori/ter ratio of *V. natriegens* in the early exponential phase is at least four and that multifork replication occurs.

Our marker frequency analyses throughout the growth curve yielded further results, detailed as follows:(1)Replication started very early in the lag phase. At thirty minutes after inoculation, ongoing replication was already observed, long before the start of cell division. Additionally, in *B. subtilis*, the onset of replication during the lag phase was observed after the transfer of stationary-phase cells to a fresh medium [3]. Similarly, when *Synechococcus elongatus* PCC 7942 was transferred from dark to light conditions, the chromosome number increased from 2–3 to 4–10 before the onset of phototrophic growth [57]. These results obtained with species of three different phylogenetic groups indicate that this strategy is widespread; however, studies with other species are needed.(2)The replication of chromosome 1 started earlier (0.5 h) than that of chromosome 2 (1 h) (Figure 5). Although this has not yet been described for *V. natriegens*, it is in line with earlier results obtained for *V. cholerae* showing that ongoing replication of chromosome 1 licenses the replication initiation of chromosome 2 [51,58,59].(3)For *V. cholerae*, it has been shown that only the initiation of replication of the two chromosomes occurs at different times; in contrast, the termination of replication occurs simultaneously and is coordinated with cell division [50]. Our results indicate that this might also be true for *V. natriegens* because for both chromosomes, there was a slight difference in the read numbers at 4 h, but replication had completely ceased at 5 h.(4)The ori/ter ratio was much higher for chromosome 1 than for chromosome 2 (Figure 5). This was also observed in a recent study on *V. natriegens* in the early exponential phase, which reported an ori/ter ratio of 5 for chromosome 1 and 2.5 for chromosome 2 [56]. This was due to the following two reasons: first, the origin copy number increase is much higher for chromosome 1 than for chromosome 2 (Figure 2), and second, the difference depends on the size of the chromosome because larger chromosomes require a longer time between initiation and termination.(5)The number of origins and the ori/ter ratio had already declined during the exponential growth phase. In contrast, it is thought that the growth rate, number of origins, and ori/ter ratio are constant during fast exponential growth for the mero-oligoploid species *E. coli* and *B. subtilis*. However, quantification of the numbers of origins and termini during the whole growth curve, as presented herein, is missing for these two intensively studied model species.

The finding that *V. natriegens* is mero-oligoploid during the lag phase and exponential phase was unexpected because *V. cholerae*, another species of the same genus, is monoploid. *V. cholerae* has been studied very intensively; in particular, its origin regions have been labeled with fluorescent proteins in various studies. Therefore, the origins and their dynamics in the cell cycle can be directly observed as fluorescent foci under a microscope [49,50,51,59,60,61,62,63]. The largest fraction of cells contained one origin (before the initiation of replication) or two origins (after the initiation of replication) for both chromosomes. For example, cells growing exponentially in LB medium at 30 °C were analyzed at an OD _600_ of 0.15, and 33% exhibited one fluorescent focus, and 66% had two fluorescent foci for chromosome 1, with values of 66% and 34% for chromosome 2 [50]. Similar results have been observed in other studies. Several explanations seem possible for the striking difference between the 1–2 origins observed for *V. cholerae* in various studies and the up to 50 origins for *V. natriegens* observed in the current study:(1)The different results may have been caused by differences in the applied conditions, e.g., temperature, medium, and aeration. For example, marker frequency analyses of exponentially growing *Vibrio parahaemolyticus* cultures revealed an ori/ter ratio of 4.0 for growth in a complex medium but only 1.4 in a synthetic medium [58]. Similarly, fast-growing *E. coli* cells contain on average 6.5/6.8 origins, and slow-growing cultures contain only 2.0/2.5 origins [1,13]. As discussed above, in general, fast-growing cultures need to employ multifork replication, whereas slow-growing cultures do not. Notably, these general requirements cannot explain the high copy number of approximately 50 origins at the onset of the exponential growth phase in *V. natriegens.* Nevertheless, small changes in medium composition can have a very dramatic effect, e.g., a change in medium composition increased toxin production per cell in *V. cholerae* by more than 1000-fold [64].(2)For *V. natriegens*, the results were very sensitive to the exact point of the growth curve; for example, we measured 50 origin copies in the early exponential growth phase, 20 in the mid-exponential growth phase only 1 hour later, and only in the transition phase 2.5 h later (Figure 2). One publication about *V. cholerae* exists that contradicts all other studies, reporting a highly dynamic change in the origin copy number during the growth curve [52]. Using the qPCR approach, 10–15 origin copies were observed during mid-exponential growth, but 70 origin copies were observed during the transition phase. Therefore, specific conditions and/or strains (see below) exist that also result in highly dynamic origin copy number changes in *V. cholerae*, albeit in a completely different manner, as observed in this study on *V. natriegens.*(3)Another possible reason is species-specific biological differences in the genus *Vibrio*. For example, one study found an ori/ter ratio of 4.0 for *V. parahaemolyticus* but a ratio of only 2.0 for *V. cholerae* and *V. vulnificus* [58]. Additionally, in Cyanobacteria, species-specific differences within one genus have been reported, e.g., *Anabaena variabilis* contains 5–8 genome copies, and *A. cylindrical* contains 25 [2].(4)Another possible explanation might be strain-specific differences within one species. The species *V. cholerae* comprises several serotypes as well as strains within these serotypes [65]. A comparison of clinical and environmental isolates revealed that genetic variation exists [66]. However, these strain comparisons did not include quantification of ploidy level. Nevertheless, this might possibly explain the values of 1–2 origins and up to 70 origins for *V. cholerae* reported in different studies (see above).

The high ori/ter ratios of 4–6 at the end of the lag phase and beginning of the exponential phase also indicate that during this time, the gene dosage of origin-adjacent genes is several-fold higher than that of terminus-adjacent genes. This would be of advantage if genes with specific importance for central metabolism and fast growth are enriched near the origin. A recent bioinformatics study analyzed the genomes of 124 species of Vibrionaceae and reconstructed the pangenome. The 61,512 clusters of orthologous genes were divided into 4 groups: core genes (present in all 124 genomes), softcore genes (present in at least 117 genomes), shell genes (present in at least 3 genomes), and cloud genes (present in 1 or 2 genomes). For *V. natriegens* and other species, it was revealed that these gene groups have an unequal distribution (Figure 7, taken from [67]): (1) core and softcore genes are enriched on chromosome 1 and only scarcely found on chromosome 2, and (2) core and softcore genes on chromosome 1 are enriched in the origin-adjacent genome. In stark contrast, shell and cloud genes are enriched on chromosome 2, and on chromosome 1, they are enriched in the terminus-adjacent half. It was hypothesized that this unequal distribution is due to specific intracellular localizations of different parts of the chromosome. However, our results suggest a different evolutionary driving force for this observed unequal distribution, i.e., that core and softcore genes encode central functions that are important for fast growth, with the greatest benefit of a high gene dosage near the origin of chromosome 1.

For *V. natriegens*, this is a hypothesis; however, for other species, experimental evidence is available of origin-adjacent localizations of specific genes correlating with a growth advantage. One example is the S10 operon of *V. cholerae*, which encodes ribosomal proteins and is located near the origin of chromosome 1. Several mutants have been constructed in which the S10 operon is moved 35 kb, 510 kb, or 1120 kb in the direction of terminus 1 or in which it is moved from a position near the origin of chromosome 1 to a position near the terminus of chromosome 2 [68]. In these mutants, the S10/ter ratio and the relative S10 expression became increasingly reduced. Importantly, the growth rates of the mutant strains were also increasingly reduced to a level of approximately 15% in the strain with the S10 operon near the terminus of chromosome 2. In addition, the efficiency of *V. cholerae* infecting *Drosophila melanogaster* was decreased by approximately three orders of magnitude in the strain with S10 near the terminus of chromosome 2 compared with a wild-type strain containing S10 near the origin of chromosome 1 [68]. In a follow-up study, four isogenic strains with the S10 operon at different positions on chromosome 1 were subjected to a laboratory evolution experiment [69]. The aim was to evolve faster-growing strains, and indeed, all strains showed increased growth rates due to various mutations. However, even after 1000 generations, secondary mutations could not compensate for the growth advantage of the strains with the S10 operon near the origin [69].

The genomic location of genes can also influence processes other than the growth rate. For example, two isogenic *B. subtilis* mutants with sporulation network genes close to the origin or the terminus exhibit different sporulation patterns [70]. In *Bacillus cereus*, *Staphylococcus aureus*, and *E. coli*, treatment with antibiotics that target replication increases the ori/ter ratio 2–5-fold and leads to the increased expression of near-origin genes [71]. It was suggested that a near-origin location of DNA repair genes would allow for an automatic and robust response to replication stress. Several biological processes that are influenced by the chromosomal locations of genes have been recently reviewed [72].

Although different evolutionary advantages of a gene location near the replication origin might exist, the most parsimonious explanation for the outgrowth of *V. natriegens* after switching from famine to feast conditions is the rapid increase in the gene dosage of origin-adjacent genes to enable the rapid cell volume increase and the onset of exponential growth that we observed. The advantage of a near-origin location would be considerably higher for *V. natriegens* than for *V. cholerae* due to the higher ori/ter ratio. This can and should be tested in future projects of groups working with *V. natriegens*.

## 5. Conclusions

In this study, the copy numbers of sites near the replication origin and the replication terminus (and two additional sites) of both chromosomes of *V. natriegens* were quantified using qPCR. As an independent approach, marker frequency analyses were also performed. Importantly, nine time points throughout the growth curve were analyzed with both methods, from the lag phase throughout exponential growth and the transition phase to the stationary phase. Rapid and dynamic regulation of replication was observed, with a high increase in the ori/ter ratio of chromosome 1 as well as the absolute gene dosage of origin 1. This dynamic regulation of replication was accompanied by a large cell volume increase during the lag phase. During exponential growth, the ori/ter ratios of both chromosomes and the cell volume decreased. This dynamic regulation is different from all other species that have been previously characterized, including the close relative *V. cholerae*. These results will help to guide the design of future studies in basic molecular genetic research as well as the biotechnological applications of the fastest-growing bacterium, which has rapidly gained attention in recent years.

## Figures and Tables

**Figure 1 genes-14-01437-f001:**
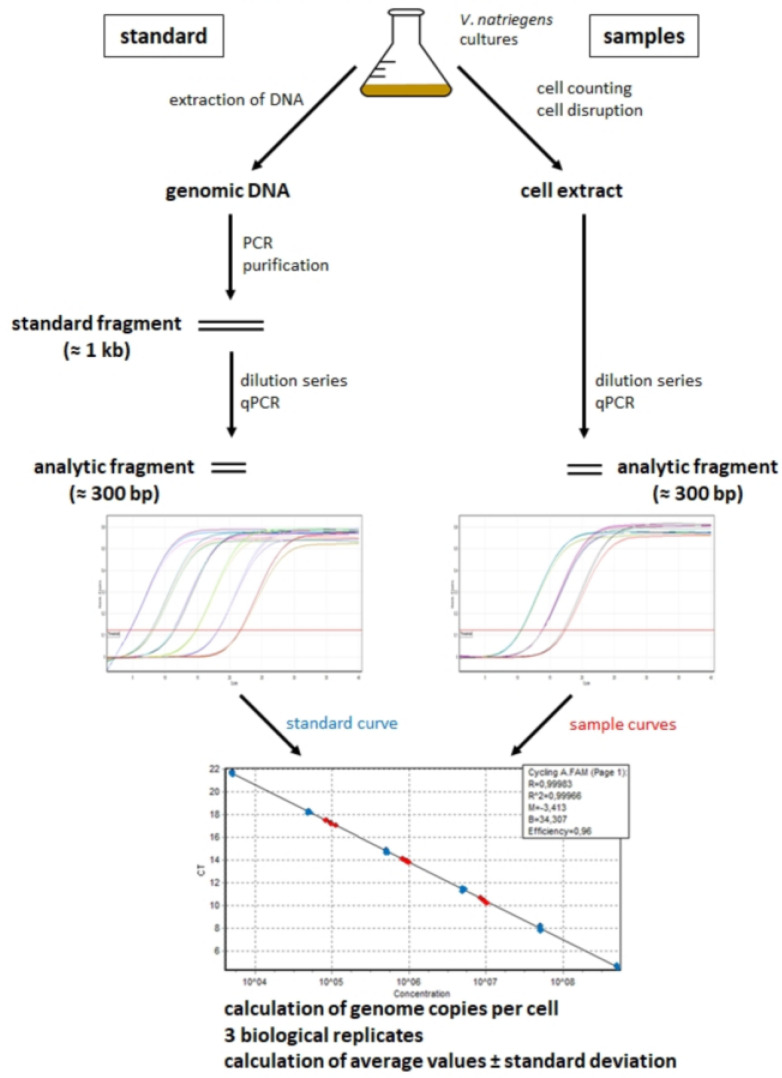
Schematic overview of the real-time PCR method for quantification of chromosome copy numbers per cell. Note that the method enables quantification of different selected sites on several replicons (taken from [14], with modifications).

**Figure 2 genes-14-01437-f002:**
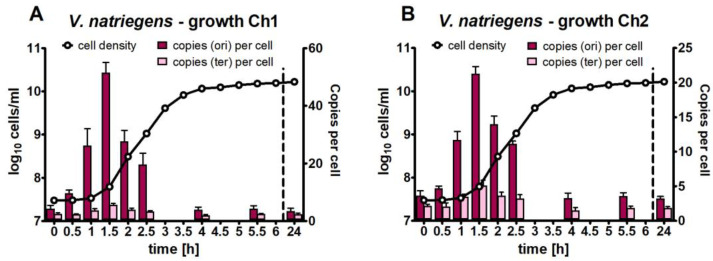
Quantification of the origin and terminus copy numbers of chromosomes 1 (**A**) and 2 (**B**) of *V. natriegens* throughout the growth curve. The left *y*-axis shows the cell densities (logarithmic scale), and the right *y*-axis shows the copy number per cell. The origin copy numbers are shown in dark red, and the terminus copy numbers are shown in pink. Average values and their standard deviations of three biological replicates are shown.

**Figure 3 genes-14-01437-f003:**
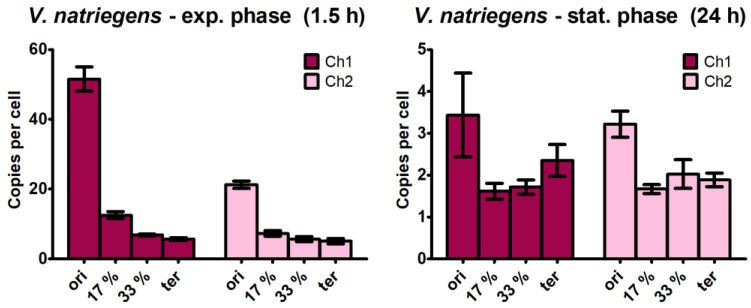
Quantification of four sites of chromosomes 1 and 2 at the early exponential phase (1.5 h post-inoculation) and late stationary phase (24 h post-inoculation). The results for chromosome 1 are shown in dark red, and the results for chromosome 2 are shown in pink. Average values and their standard deviations of three biological replicates are shown.

**Figure 4 genes-14-01437-f004:**
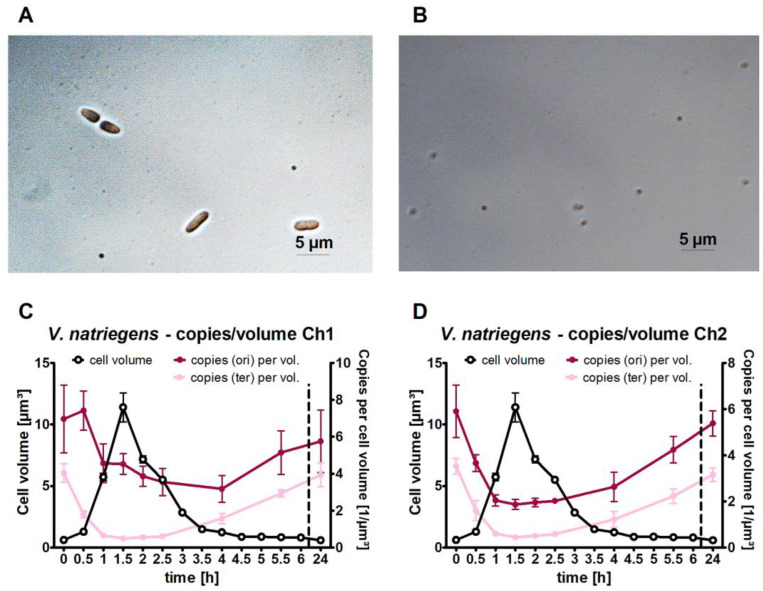
Dynamic changes in the cell volume of *V. natriegens* and the origin/terminus copy number per unit cell volume. (**A**,**B**) show microscopic pictures of cells in the early exponential phase (1.5 h post-inoculation) and in the late stationary phase (24 h), respectively. Contrast enhancement was used to optimize the visualization of the cells. The copy numbers of the origins and termini per unit cell volume are shown for chromosome 1 (**C**) and chromosome 2 (**D**) throughout the growth curve. The left *y*-axis shows the average cell volume, and the right *y*-axis shows the origin (dark red) and terminus (pink) copy numbers per unit cell volume. Average values of three biological replicates and their standard deviation are shown.

**Figure 5 genes-14-01437-f005:**
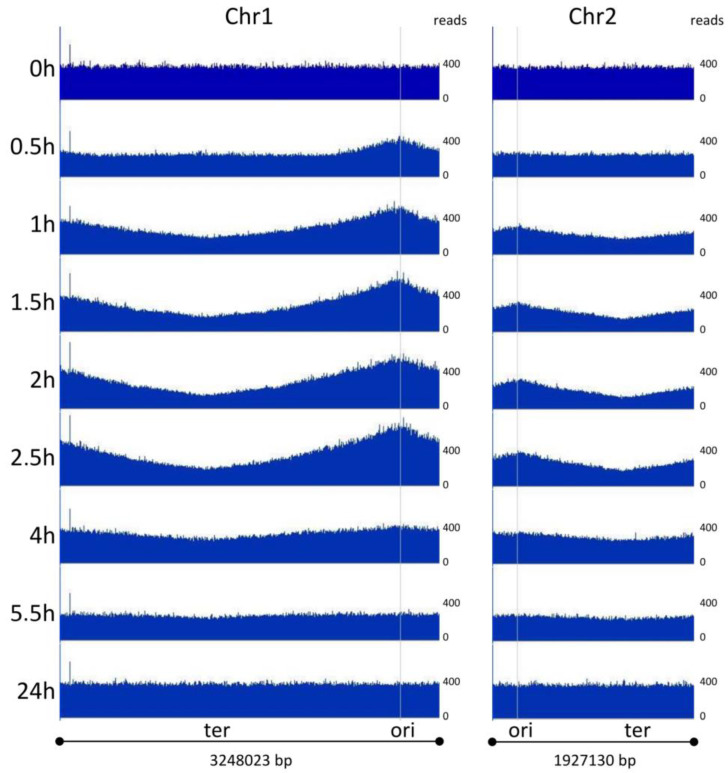
Marker frequency analyses of both chromosomes of *V. natriegens* at nine time points throughout the growth curve. The time points are indicated to the left and are identical to the time points of the qPCR analyses. For each of the nine time points, next-generation sequencing generated between seven and ten million reads, which were mapped onto the two chromosomes. The number of reads obtained for every genomic site is shown without normalization. The locations of the origins and the termini of the two chromosomes are marked.

**Figure 6 genes-14-01437-f006:**
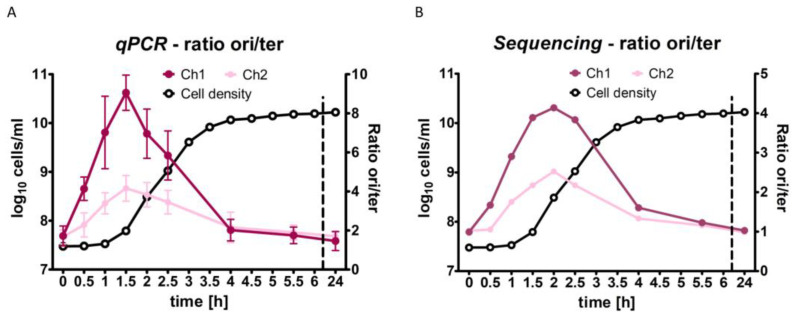
The ori/ter ratios calculated for the nine time points throughout the growth curve are shown (**right**
*y*-axis). The results for chromosome 1 are shown in dark red, and the results for chromosome 2 are shown in pink. The growth curve is shown in black (**left**
*y*-axis). (**A**) The ori/ter ratios obtained with the qPCR method. (**B**) The ori/ter ratios obtained with marker frequency analysis.

**Figure 7 genes-14-01437-f007:**
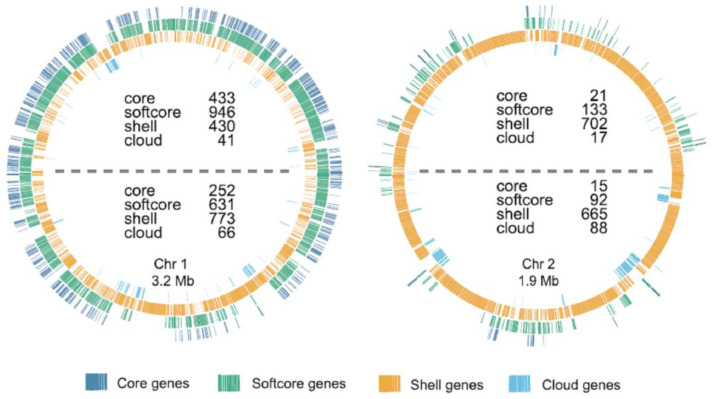
Maps of chromosomes 1 and 2 of *V. natriegens*. The distributions of core genes (dark blue), softcore genes (green), shell genes (yellow), and cloud genes (light blue) are indicated in four rings from outermost to innermost. The figure is taken from [67]. The origins of both chromosomes are at the top of the respective circle, and the termini are at the bottom.

**Table 1 genes-14-01437-t001:** Compilation of the origin and terminus copy numbers of both chromosomes in the early exponential phase and late stationary phase.

Time Point	Average Origin Copy No.	Avg. Terminus Copy No.	Ratio Origin No./Terminus No.
Chromosome 1			
1.5 h	51.6 ± 3.5	5.7 ± 0.4	9.0
24 h	3.4 ± 1.0	2.3 ± 0.4	1.4
Chromosome 2			
1.5 h	21.3 ± 1.0	5.1 ± 0.8	4.1
24 h	3.2 ± 0.3	1.9 ± 0.2	1.6

## Data Availability

The raw next-generation sequencing results and the processed results are available at the Galaxy Web Server: https://usegalaxy.eu/u/andreas_borst/h/vibriogens-brck.

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
