# Peer review of "Ploidy in Vibrio natriegens: Very Dynamic and Rapidly Changing Copy Numbers of Both Chromosomes"

_genes, 2023, doi:10.3390/genes14071437_

Round 1

Reviewer 1 Report

  Vibrio natriegens is a microorganism with the fastest-growth, special characteristics, and application potential. It still has many unknown aspects, such as why it grows so fast, how to improve cell density, etc., as well as the ploidy level and cell volume studies in this manuscript. Therefore, qPCR was used to investigate the chromosome copy numbers of four loci (origin, the 17% and 33% loci of chromosome from the origin, and the terminus) at different growth stages, and the cell volume was calculated by microscope. The results showed that the copy numbers of the two origins increased rapidly at the end of the lag phase, reached the highest at the beginning of the exponential phase (1.5 h), and then decreased sharply and finally stayed at ~3. On the contrary, the copy numbers of the other 3 chromosome loci did not vary significantly during the whole growth process and remained at low levels. Monitoring cell volumes during growth showed similar results, in which cell volumes increased rapidly at the end of the lag phase, reached a maximum at 1.5 h, and decreased rapidly thereafter.

    The experimental methods and ideas adopted in this manuscript were the same as the previously reported paper about the Vibrio cholerae ploidy research (doi: 10.1093/femsle/fnx190), so the research methods were not innovative enough. From the experimental results obtained in this work, the authors speculated that the core genes that enable the fast growth of V. natriegens may locate near the two origins. If the authors could further prove this hypothesis through genetics or other methods, the importance of this manuscript would be revealed.

Comments:

1. It is suggested that the authors should further explore which core genes affect the growth rate of V. natriegens near the two replicons through genetics or other methods.

2. Line 331: As mentioned by the authors, the variation trends of B. subtilis chromosome copy number is like that of V. natriegens. However, why is the growth rate of B. subtilis much lower than that of V. natriegens? They should have a full discussion.

3. Line 246: why did the authors choose the two genome loci (17% and 33% from the origins)? It is suggested to give reasons.

4. It is suggested that the authors should explicitly state that cell volume was getting bigger/smaller, and chromosome copy number was getting higher/lower, rather than simply using "vary" to summarize in the Abstract section.

5. Grammatical errors:

Line 27: “However, during the last decade it has been revealed that many if not the majority of prokaryotic species are oligoploid (2 – 10 copies)”.

Line 63: “because it can take up exogenous DNA and incorporate it into its genome”.

Line 249: “If replication would be in steady state and would proceed with linear speed, copy numbers of 1/3 and 2/3, respectively, of the copy number differences between origins and termini would be expected”.

Line 319: “It has been reported that the copy chromosomal copy number…”.

6. The full name should be given with some abbreviations, such as line 74: DOPA.

Author Response

Thank you very much for the time you invested to review our manuscript and for your comments. First, I would like to answer to your general comments, and then answer your six specific comments.

- The quality of the English language was rated totally differently by the three reviewers. I do not want to start any discussion, but the language will be given to the “American Journal Experts” for language editing after all three reviewers agreed with the scientific content of the manuscript.

- The methods were not adapted from a Vibrio cholerae paper, but were introduced and optimized by us in 2006 as we quantified the genome copy numbers in halophilic archaea (PMID 17183724). Since then we have applied the qPCR method for the quantification of genome copy numbers in various phylogenetic groups of prokaryotes (e.g. PMID 21305010, 21097629, 25338080, 26919857, 29315386). We have never claimed that the method is very “innovative”, it is just very informative.

- In the first version the Discussion was rather short. Because all three reviewers proposed to discuss additional points (albeit not the same topics) and to cite many additional papers, the Discussion hat been completely re-written.

Specific comments

1) The analysis of core genes near the two origins of replications of the two chromosomes with “genetic methods” would be a complete new and additional project that would require at least one PhD thesis. The proposal that core genes are enriched near the origins is not part of the results section of our manuscript, but was observed and published by a bioinformatics group, and this paper was cited in the Discussion. We have added citations to various papers showing that important genes are often near the replication origin in different species of bacteria, including Vibrio cholerae. We have also added an additional experiment (proposed by reviewer 3) that underscores our results obtained with qPCR, i.e. marker frequency analyses at nine time points along the growth curve.

2) The growth rates of V. natriegens with 14 min. is not very different from that of B. subtilis with 24 min. The important point is that both species (as well as fast growing E. coli) have a doubling time that is shorter than the time for replication and segregation of the chromosome (about 60 min.). Under such conditions fast-growing species switch to multi-fork replication, and this phenomenon is now explained and discussed in greater detail. The faster the growth of a specific species, the higher the average number of origins has to be, and this is true for V. natriegens and B. subtilis (more than 20 origin copies versus 6 origin copies)

3) The four sites with 0%, 17%, 33% and 50% are equidistant on one arm of the chromosome, and this is now explained in more detail in the manuscript.

4) The Abstract has been changed according to this suggestion.

5) As stated above, after the approval of all three reviewers for the scientific content the manuscript will be given to language editing before publication.

6) l-DOPA is not an abbreviation, but the name of a substance. It is a precursor of several neurotransmitters and therefore of medical relevance. The alternative name levodopa has now been included.

Reviewer 2 Report

The authors investigate the chromosome copy number in Vibrio natriegens, a promising synthetic biology and biotechnology chassis. They use qRTPCR on cells all along the growth curve and investigate sites on both chromosomes. The authors find that Vnat's Chr1 origin reaches a high copy number leading up to log phase, after which copy number drops. Similarly with Chr2. The ratio of origin to terminus copy number is higher for Chr1 than Chr2 at the onset of log phase. The study also finds that the origin in particular, and not sites at the 17% or 33% positions are at the higher copy number, indicating that the replication of chromosomes in Vnat is not steady along their length. The authors discuss how this is a unique trait that has not been observed before and briefly mention evolutionary implications for V. natriegens.

The findings are interesting and lead to further questions about gene copy numbers in V. natriegens.

Comments/concerns:

1. This is an interesting paper and provides a nice insight into chromosome copy number in Vibrio natriegens.  The group has a track record of investigating ploidy in various organisms and clearly knows the field and methods well. To be more relevant to researchers who work with V. natriegens, please take note of the comments below.

2. The whole paper needs to be carefully edited for grammar and spelling. Especially the third paragraph, lines 53-74, has many issues. But most others do as well. Please check for the correct spelling of the various organisms you mention (ie. Vibrio harveyi not harvey etc). The use of words such as "very" needs to be limited, and please use more concrete phrases (x-fold, y orders of magnitude) instead of "much higher" etc.

3. There are several statements that are either incorrect or misleading. First, though 10 minute doubling time for V. natriegens was reported, it has not been seen in recent papers. Hence a statement such as " ... reported doubling time of 10 min" is more standard. Most researchers can reach 12-14 min doubling time realistically. Additionally the reason for fast growth in V. natriegens has not been experimentally identified - please change the statement to read that the rRNA operon copy number COULD be a reason.

3. It would be useful to show better quality microscopy pictures, especially for 4B, and describe in detail in the methods section how the volume was calculated from the pictures. Please be sure to state if this is a standard method and provide citations. Vibrio natriegens cells look almost like a sphere when in stationary phase, so please check if this is not a more appropriate way to calculate their volume at that stage.

4. In Figure 6 and and the paragraph describing, please define core vs softcore vs shell vs cloud.  Also please label the origin, 17%, 33%, and terminus on the chromosomes in Figure 6.

5. It may also be helpful to provide a table of sorts for the investigated timepoints of the resultant copy numbers of origin, 17%, 33%, and terminus, to summarize. 

6. In your last paragraph of the discussion you propose that "the genes encoding proteins important for fast growth were enriched near the origin". If there is indeed some evidence for this, please elaborate on this in the discussion. What are the genes around the origin that could contribute to fast growth, and how?  Is there evidence in other organisms of this? Please expand on evidence in other organisms for using chromosomal localization of genes to regulate global gene expression in the way that you mention.

7. Overall you have a unique observation about V. natriegens that may contribute to our understanding of its fast growth and metabolism. So adding a few more sentences with regards to that would make it much more relevant to the Vibrio natriegens community.

Author Response

Thank you very much for the time you invested to review our manuscript and for your comments.

In the first version the Discussion was rather short. Because all three reviewers proposed to discuss additional points (albeit not the same topics) and to cite many additional papers, the Discussion hat been completely re-written.

Here are the answers to your specific comments:

1) Thank you for this nice comment.

2) The quality of the English language was rated totally differently by the three reviewers. I do not want to start any discussion, but the language will be given to the “American Journal Experts” for language editing after all three reviewers agreed with the scientific content of the manuscript. I have already looked for typos in species names and reduced the usage of “very” in the manuscript.

3) Thank you for this suggestion. I have looked up the original paper that claimed that the doubling time is below 10 minutes and realized that this was only true for the 15 minutes of fastest exponential growth. This is now discussed in the Introduction. The high number of rRNA operons is now discussed as one POSSIBLE explanation.

Another 3) Unfortunately, we do not have better quality pictures, but I think that the dramatic difference in cell size is clearly visible. If the microscopic pictures would not be good enough, I would remove them and we would go without the visualization at two time points. The mathematical formula for a cylinder has been added (V = π r2 h). It is true that in stationary phase the cells get much shorter (as visible in the two pictures). If we would use the mathematical formula of a sphere (V = 4/3 π r3) instead of that of a cylinder, the difference would be just 30% and the large difference in cell size would remain. Because even in later growth phases many cells are more short rods than true spheres, we used the same formula for all nine time points.

4) We have added definitions for the four groups of genes. Figure six was taken from Sonnenberg et al., therefore, I do not like to change it. But I have added to the Figure legend that the origins are on the top and the termini at the bottom of the two circles.

5) I think that a Figure gives a much better overview of the results as a Table would. Therefore, I have not replaced Figure 3 with a Table. The values at the 17% and 33% sites were not determined at all nine time points, but only at the two time points shown in Figure 2. However, we have added another experiment, i.e. we performed marker frequency analyses at all nine time points. Therefore, the relative frequencies of every bases pair of both chromosomes are now shown for all nine time points.

6) Thank you for this suggestion. We have added various papers with experimental evidence for the importance of origin-near genes in V. cholerae and several other bacteria.

7) We have elaborated on the fact that our study is the first and only one that quantified the copy number of both chromosomes not only during exponential growth, but at nine time points throughout the whole growth curve.

Reviewer 3 Report

This paper describes the replication dynamics along the growth phases in Vibrio natriegens. Like all vibrios, this bacterium has a bipartite chromsome. To follow the replication dynamics, the authors mainly use the replication origin of the main chromosome (ori1), the ori2 terminal region of chromosome 1 (ter1) and ter2 specific primers (and some other markers) for quantitative PCR (qPCR) assays. They also quantify cell number and size. They show that V. natriegens is polyploid and that ploidy varies along the growth curve. They conclude that core genes are located near the origin of replication. 

Major points:

1) Overall, these findings are difficult to reconcile with well-established studies of bacterial chromosomes, which are generally haploid except for a few specific species (cyanobacteria and giant bacteria such as Epulopiscium). In particular, it is well established that V. cholerae is haploid. In L. 79, the authors suggest that V. cholerae is not haploid but polyploid. Most studies to date indicate that V. cholerae is not polyploid. The cited paper (/10.1093/femsle/fnx190) uses an indirect method to calculate ploidy. There are many studies using direct methods (e.g. FROS, flow citometry, biochemistry) showing that there is only one copy per cell (from 4 different labs in different countries). Just a few examples:

-10.1371/journal.pgen.1004557

-10.1038/nmicrobiol.2016.94

-10.1371/journal.pgen.1002472

-10.1128/JB.01067-10

-10.1128/JB.00362-07

-10.1128/JB.188.3.1060-1070.2006

-10.1371/journal.pgen.1007426

-10.1111/j.1365-2958.2004.04379.x

In fact, the chromosome stoichiometry has been successfully altered artificially:

-10.1073/pnas.0608341104

-10.1126/sciadv.1501914

Can the authors comment on these differences? Is there perhaps a synthesis between the authors' observations and current Vibrio models? 

2) One of the main findings of the authors is that V. natriegens is polyploid. While this is not impossible, it would be very provocative. It would be very different from a well-characterized related species such as V. cholerae. This conclusion may be wrong if there are some methodological errors. The latter may arise from measuring the ori per cell in an indirect way.  There could be errors either in the absolute quantification of DNA by qPCR (which seems less likely, although Nanodrop may underestimate DNA mass at low concentrations) or in the determination of cell numbers. In fact, for an organism with a doubling time of 10 minutes, there could be a problem with protocols that involve multiple steps, such as assessing cell numbers in a Neubauer chamber.  Since it involves several steps to get the measurement, it will have a large intrinsic error. However, since my opinion and the authors' findings are hard to concile, the best choice is to provide different independent methods to make their point. One possible choice is to apply Fluorescence Repressor Operator System (FROS) targeting a region close to ter. Polyploidy should appear as multiple foci within the same cell. Alternatively, FISH could be performed. Another choice could measure cell size and DNA content using flow cytometry coupled to DAPI staining. This could provide additional information that could support the author's hypothesis.

3)L.85 ori1/ter1 ratio has already been established in exponential phase for V. natriegens using Marker Frequency Analysis.  

-10.1186/s12864-022-08831-y

Indeed, this is a well conserved trait, as we have known for more than a decade, as shown in this beautiful (and often neglected!) paper from the Tetsuda Iida lab (Dryselius et al 2008, 10.1186/1471-2164-9-559).

The novelty of the authors' work is to provide this ratio at different moments of the growth curves. This should be emphasized. 

4) As mentioned above, for an organism with a doubling time of 10 minutes, there may be a problem with protocols that involve multiple steps, such as counting cells in a Neubauer chamber or measuring cell size on the microscope. Since there are several steps to get the measurement, they will have a large intrinsic error. Meanwhile, the number of cells treated by microcopy (25 per sample) seems a bit low by today's standards. May I suggest to analyze the images with MicrobeJ (10.1038/nmicrobiol.2016.77)?

5)L321. There are many studies showing the original ratio of both chromosomes of various vibrios, especially V. cholerae. The authors cite only one paper (which is different from the rest of the bibliography). Therefore, I think that some of the following comments may enrich the discussion: 

-10.1186/1471-2164-9-559 ori/ter of several Vibrio species

-10.1126/sciadv.1501914 Marker Frequency Analysis of V. cholerae

-10.1371/journal.pgen.1005156 qPCR ori/ter of both chromosomes

-10.1186/s12915-020-00777-5 Marker frequency analysis of V. cholerae 

-10.1128/mBio.00097-17 Marker Frequency Analysis of V. cholerae different media

-10.1186/s12864-022-08831-y Marker frequency analysis of V. natriegens

The authors should emphasize that they took measures along the entire growth curve, as this is their most original contribution. 

Minor points: 

6) Abstract needs to be nuanced (cell size varies "enormously") and jargon should be avoided ("shell genes"). 

7) The first 10-15 citations should be self-citations. Are there other authors working on polyploidy in bacteria?  

8)L340. The discussion of this paper seems very appropriate, but first discuss or explain the terms "core", "soft core", "cloud" and "shell" genes.

9)Figure 2 is very interesting. It also shows the coordination of replication between chromosome 1 and 2, since chromosome 2 starts in all vibrios studied after 2/3 of the main chromosome has occurred. I also suggest adding a third panel plotting ori1/ter1 over time and along OD. This will show how replication "drives" the growth curve. 

10) Discuss these results in light of marker frequency analysis and ori/ter studies already done in other vibrios. Statistical analysis is lacking.

 11) The direct correlation between cell size and growth rate is an old observation from the Copenhagen School (Schaechter's growth law). You may want to cite a reference such as 10.1088/1361-6633/aaa628; 

12) L. 352 This is the case for many genes where this has already been elegantly demonstrated. You may want to mention 10.1016/j.tim.2016.06.003 and references therein such as:-10.1016/j.cell.2015.06.012

-10.1016/j.cell.2014.01.068 

-10.1093/nar/gkv709

-10.1371/journal.pgen.1005156

-10.1371/journal.pgen.1005156 

13) The concept of "mero-polyploidy" is less informative than multi-fork replication or genome-wide copy number (/10.1016/j.tim.2016.06.003). The former overlooks that the excess of copies near the ori is the result of the replisome not being fast enough to duplicate the entire genome faster than the doubling time. As a result, the cell overlaps replication rounds within the same cell cycle. This phenomenon needs to be explained somewhere in the manuscript.

14)L.54 Vibrio natriegens was first described as Pseudomonas. The first papers date from the late 50's, but the paper that first described its particularly rapid growth is "

PSEUDOMONAS NATRIEGENS, A MARINE BACTERIUM WITH A GENERATION TIME OF LESS THAN 10 MINUTES" by RG Eagon - J. Bact., 1962.

15)L.63. Here it is really necessary to mention the very first person who showed that 

I-among Vibrios V. natriegens could be induced for natural competence for transformation, in particular Dalia lab work: /10.1021/acssynbio.7b00116

ii- the first to establish molecular biology tools for V. natriegens (/10.1038/nmeth.3970), in particular CRISPR-Cas9 tools (/10.1038/s41564-019-0423-8).

Author Response

Thank you very much for this very detailed review and the large number of suggested papers that we should look at. The review was very helpful and clearly improved the quality of the manuscript considerably.

The quality of the English language was rated totally differently by the three reviewers. I do not want to start any discussion, but the language will be given to the “American Journal Experts” for language editing after all three reviewers agreed with the scientific content of the manuscript.

In the first version the Discussion was rather short. Because all three reviewers proposed to discuss additional points (albeit not the same topics) and to cite many additional papers, the Discussion hat been completely re-written.

Here are the answers to the specific comments:

1) During the last decade we have quantified the genome copy numbers in various phylogenetic groups of bacteria and archaea. Obviously, the current project was the first time that we worked with Vibrio, and we are no experts for this genus. I did perform a literature search for Vibrio natriegens, but, unfortunately, failed to do so for V. cholerae. Thank you for pointing out that the one V. cholerae  paper that I had found by chance is totally untypical and that many studies showed that the species is monoploid. The discussion has been completely rewritten. I included many of the suggested references and I included a discussion about the possible reasons for the differences between the large body of literature about V. cholerae  and our results with V. natriegens.

2) We have followed the suggestion to reinforce our results by using a second, independent experimental approach. We have performed marker frequency analyses for all nine time points of the growth curve that had been analyzed with the qPCR method. The results nicely underscore our earlier results and also show that the origin copy number is regulated very dynamically, that upregulation starts early in lag phase, etc.

We do not think that counting the cell numbers with a Neubauer chamber has “ large intrinsic error”. Under optimal conditions (37oC, good aeration) the cells divide every 14 minutes, after removal of a sample (room temperature, no aeration) cell division ceases. Counting of a sample takes less than 5 minutes. We have made the test to remove a sample from an exponentially growing culture, count it directly, and let it stand at room temperature, and count it again. The results were identical.

3) Thank you, we have integrated the two papers. I have tried to emphasize more that this study for the first time analyzed V. natriegens at nine time points throughout the whole growth curve.

4) As argued above, cell counting with a Neubauer chamber does not include a large intrinsic error. Thank you for pointing MicrobeJ out to us, we will use it in future studies. Unfortunately, cell volumes were determined live at the screen, and not using saved pictures. Therefore, we only have pictures for the two time points shown in the manuscript to visualize the large cell size differences. We agree that the number of 25 cells is rather low, however, the variances (standard deviations) are included in Figure 4 and they are rather or very small. The more than tenfold difference in average cell volume is clearly visible.

5) The results for V. cholerae  and other Vibrio species have been integrated. Thank you again for suggesting relevant references.

6) The abstract has been revised accordingly.

7) Indeed, 8 of the first 15 papers are self-citations, 7 are citations of papers from various other groups. In fact, only two groups worldwide systematically work on ploidy in prokaryotes, that is the group of Esther Angert in the USA (who works exclusively on giant bacteria) and my group. In every first paper of a new phylogenetic group we have included a Table containing all pre-published results for that group and give references.

8) The terms have been explained.

9) A Figure with the ori/term ratios along the growth curve is included in the Discussion, both for the qPCR results and the marker frequency analyses results.

10) Marker frequency analyses with other Vibrio species have been included. Standard deviations are reported for all analyses except the marker frequency analysis, which were performed only once.

11) We do not feel that the Copenhagen School and Schaechter’s growth law are relevant in this case, because they claim a constant volume during steady state exponential growth, while, in contrast, we see dynamic changes during lag phase and during exponential growth.

12) Thank you for pointing out these references. The results and the references have been included.

13) In my opinion “multi-fork replication” and “mero-oligoploidy” are redundant terms to point out the phenomenon that replication cycles are intertwined in fast-growing bacteria. A detailed explanation of multifork-replication has been added.

14) The paper that officially introduced the novel species Pseudomonas natriegens has been added.

15) The two suggested papers have been added.

Round 2

Reviewer 3 Report

Dear Authors,

The Editorial Office gave me 3 days to review your paper, and I would have liked to have some more time to thoroughly review your work given the  effort you have put into it. First, let me commend you on the significant improvement in your work. Congratulations on that. I believe that the MFA analysis has taken your work to another level and can potentially provide further information (which you can utilize and employ in a follow-up study, e.g. consider the slope of the plot in log phase ). However, I must reiterate my skepticism about polyploidy in Vibrio natriegens. Nevertheless, it is important to let the scientific community judge this aspect over time. Overall, I believe you have adequately addressed most of my concerns. However, I would like to suggest considering the experiments I proposed in point #2. For instance, applying the Fluorescence Repressor Operator System (FROS) targeting a region close to ter could reveal polyploidy as multiple foci within the same cell. Alternatively, performing FISH or measuring cell size and DNA content using flow cytometry coupled with DAPI staining could provide additional information to support the authors' hypothesis. These experiments might prove useful in future work.

Additionally, I have observed a significant improvement in the discussion and introduction sections, which has enhanced the overall quality of the work. I did notice a few remaining typos, but I believe those can be addressed during the post-production phase. Our job in the technical aspects is done.

Once again, congratulations to the authors.

Best regards,

Author Response

Dear reviewer,

thank you very much for your very positive reaction to the revised version of our manuscript.

Of course I agree that a microscopy picture with many foci per cell would nicely visualise the fact that the cell contains many origins (if the foci would be separate and not just one very bright spot). In fact we made one attempt to use FISH. As you know, FISH is mostly used to target ribosomes, which have a copy number of 20 000 - 70 000 per cell, and is only rarely used to detect single copy genes. To enhance the signal, we have generated two mixed probes (each composed of 8 single probes) for a region near the origin and near the terminus, respectively. Unfortunately, we could not see fluorescent foci at all, but we are not experts in microscopy. The first author left the group, so there was no chance to optimize the procedure. Therefore, the study is left with "only" two independent methods, qPCR and marker frequency analysis. This is our first and last paper with V. natriegens, but I hope that it will induce follow-up experiments by groups of the V. natriegens community.

Currently I wait for the reactions of the other two reviewers and hope that they will also be positive.

Once again, thank you very much for your time, and especially for correcting my lack of knowledge about V. cholerae. Your review enhanced the quality of the manuscript considerably.

best wishes

Jörg Soppa